# Exploring Cardiac Exosomal RNAs of Acute Myocardial Infarction

**DOI:** 10.3390/biomedicines12020430

**Published:** 2024-02-14

**Authors:** Seung Eun Jung, Sang Woo Kim, Jung-Won Choi

**Affiliations:** 1Medical Science Research Institute, College of Medicine, Catholic Kwandong University, Gangneung-si 25601, Republic of Korea; top98@naver.com; 2International St. Mary’s Hospital, Incheon 22711, Republic of Korea; ksw74@cku.ac.kr; 3Department of Convergence Science, College of Medicine, Catholic Kwandong University, Gangneung-si 25601, Republic of Korea

**Keywords:** exosomal RNA sequencing, exosome, acute myocardial infarction, microRNA, cardiac tissue

## Abstract

Background: Myocardial infarction (MI), often a frequent symptom of coronary artery disease (CAD), is a leading cause of death and disability worldwide. Acute myocardial infarction (AMI), a major form of cardiovascular disease, necessitates a deep understanding of its complex pathophysiology to develop innovative therapeutic strategies. Exosomal RNAs (exoRNA), particularly microRNAs (miRNAs) within cardiac tissues, play a critical role in intercellular communication and pathophysiological processes of AMI. Methods: This study aimed to delineate the exoRNA landscape, focusing especially on miRNAs in animal models using high-throughput sequencing. The approach included sequencing analysis to identify significant miRNAs in AMI, followed by validation of the functions of selected miRNAs through in vitro studies involving primary cardiomyocytes and fibroblasts. Results: Numerous differentially expressed miRNAs in AMI were identified using five mice per group. The functions of 20 selected miRNAs were validated through in vitro studies with primary cardiomyocytes and fibroblasts. Conclusions: This research enhances understanding of post-AMI molecular changes in cardiac tissues and investigates the potential of exoRNAs as biomarkers or therapeutic targets. These findings offer new insights into the molecular mechanisms of AMIs, paving the way for RNA-based diagnostics and therapeutics and therapies and contributing to the advancement of cardiovascular medicine.

## 1. Introduction

Myocardial infarction (MI), frequently the initial symptom of coronary artery disease (CAD), is a major contributor to global mortality and disability [1]. Defined as myocardial cell death due to ischemia, MI typically results from thrombosis triggered by rupture or erosion of atherosclerotic plaque in the coronary artery [2]. Onset is rapid, occurring within 20 min of blood supply interruption, leading to irreversible cell death within a few hours [3]. Distinguishing between acute and chronic myocardial injury is essential for early MI detection and mortality reduction [4]. Acute myocardial infarction (AMI), a prominent cardiovascular disease, poses a significant global health challenge [5,6,7]. Understanding the complex pathophysiology of AMI is vital for developing novel therapeutic strategies [8]. In this complex milieu, exosomal RNAs (exoRNAs) within cardiac tissues play a pivotal role in intercellular communication and the pathophysiological process of AMI [9].

Exosomes, small extracellular vesicles (EV), have garnered significant attention in the scientific community for their role in mediating intercellular communication [10]. These vesicles are known for transporting a diverse array of biomolecules, notably RNAs, among cells, reflecting the physiological and pathological state of their source cells [11,12]. As such, they are increasingly recognized for their potential as biomarkers in disease diagnosis and prognosis. In the specific context of AMI, exoRNAs provide a window into the molecular upheavals occurring within cardiac tissues during the disease’s acute phase [13]. Recent research has expanded the understanding of EVs, particularly exosomes and microRNAs (miRNAs or miRs), in the realm of cardiovascular health. These entities are being explored for their therapeutic potential not only in cardiovascular diseases but also cancer and neurological disorders [14,15]. For instance, miR-134 has been identified as a regulator of STAT5B function, making it a promising biomarker and therapeutic agent for breast cancer [16]. Similarly, exosome-derived miR-126 from ADSCs shows potential in reducing cardiac fibrosis and inflammation, thereby mitigating myocardial injury [17]. Moreover, miR-217-containing exosomes have emerged as significant markers in chronic heart failure by modulating cardiac fibrosis and dysfunction. Additionally, exosome-derived miR-21-3p has been identified as a key mediator in cardiac hypertrophy through its regulatory impact on sorbin and SH3 domain containing 2 (*SORBS2*) and PDZ and LIM domain 5 (*PDLIM5*) [18]. 

The role of miRNAs extends beyond just biomarkers; these small noncoding RNAs are pivotal in controlling gene expression and cellular process, impacting the pathophysiological pathways of MI both directly and indirectly [19,20]. Certain miRNAs have been implicated in inducing cardiomyocyte death through mechanisms including apoptosis, autophagy, and necroptosis, while others play a role in cardiomyocyte proliferation, repair, and intercellular interaction [20]. These insights not only enhance the understanding of MI mechanisms but also underscore the potential of miRNAs as diagnostic biomarkers for this condition. 

This study delves into the comprehensive profiling of exoRNAs isolated from cardiac tissues in MI animal models. Employing high-throughput sequencing techniques, it aimed to map the exoRNA landscape in AMI, focusing on the differentially expressed miRNAs. This approach included sequencing analysis to identify significant miRNAs in AMI, followed by validation of functions of selected miRNAs through in vitro studies involving primary cardiomyocytes and fibroblasts. This research endeavors to provide a deeper understanding of molecular alterations in cardiac tissues following AMI and explores the potential of exoRNAs as biomarkers and therapeutic targets. The findings of this study make a significant contribution to cardiovascular medicine, offering novel insights into the molecular underpinnings of AMI. This research not only enriches existing knowledge but also paves the way for the development of innovative RNA-based diagnostics and therapeutic strategies, inspiring future research in this critical area of healthcare.

## 2. Materials and Methods

### 2.1. Animals

To establish an MI mouse model, 12-week-old male C57BL/6 mice (23 ± 4 g; KOATECH, Pyeongtaek, Republic of Korea) were used. Following anesthesia via zoletil (30 mg/kg; Virbac, France) and xylazine (10 mg/kg; Bayer Korea, Ansan, Republic of Korea), the mice were airway ventilated using a ventilator (Harvard Instruments, Holliston, MA, USA). The left anterior descending (LAD) artery was ligated with 6-0 prolene suture (Ethicon, Diegem, Belgium). Subsequently, muscle and skin closure were performed with 4-0 prolene suture (Ethicon) [21]. The mice were divided into three groups: sham, MI-1day, and MI-3day, based on the duration of left anterior descending LAD artery ligation, followed by suture closure [22]. The mice were sacrificed on the designated day to procure heart tissue for analysis, ensuring simultaneous tissue collection across all groups. Ten mice were selected for each group to establish an MI animal model, and finally, five mice were randomly selected per group for further experiments.

### 2.2. Triphenyltetrazolium Chloride (TTC) Staining

Before TTC staining, the isolated heart tissue was perfused with 1X phosphate-buffer saline (PBS; Biosesang, Seongnam, Republic of Korea) several times. Heart tissues from each experimental group were subjected to TTC staining. For this, tissues were incubated in the 1% TTC (Sigma-Aldrich, St. Louis, MO, USA) solution, which was prepared by dissolving TTC in 1X PBS. The incubation was carried out at 37 °C for 1 h in a state shielded from light to prevent photo-degradation [23,24]. Post-incubation, the tissues were fixed in a 4% paraformaldehyde solution (Biosesang, Seongnam, Republic of Korea) at 4 °C for 4 h. This was followed by sectioning the tissues into 1 mm thick cross-sections for detailed examination. The stained and sectioned heart tissues were then photographed using a digital camera (DIMIS M model, Anyang, Republic of Korea) to document the results of the TTC staining process. The infarcted tissue appears white, while the viable tissue is red. This photographic evidence is crucial for visualizing and analyzing the extent of infarction in heart tissues.

### 2.3. Exosome Isolation

Heart tissues isolated from each group were dissected into 1 mm^3^ pieces using a razor. These tissue fragments were then gently rinsed with cold 1X PBS in a cell strainer fitted with a 70 μm nylon mesh (SPL, Pocheon, Republic of Korea). Five heart tissue fragments from the same group were pooled together in a 50 mL tube. The pooled heart tissues in each tube were incubated in serum-free medium supplemented with 20 mM N-2-hydroxyethylpiperazine-N-2-ethane sulfonic acid (HEPES; Thermo Fisher Scientific, Grand Island, NY, USA), at 37 °C with gentle shaking (200 rpm) for 45 min [25]. This step facilitates the release of exosomes into the medium. Following incubation, the tube was centrifuged at 3000× *g* for 15 min at 4 °C. This centrifugation step aimed to pellet intact cells and cellular debris, leaving the supernatant enriched with exosomes. Exosomes were then isolated from the supernatants using an Exo2D EV isolation kit for RNA analysis (EXOSOMEplus, Seoul, Republic of Korea), following the manufacturer’s instructions. Briefly, Exo2D reagents were added to the supernatants in a 1:5 ratio, and the mixture was incubated at 4 °C for 1 h. Subsequently, the mixture was centrifugated at 3000× *g* for 30 min at 4 °C. The resultant white pellets containing the exosomes were resuspended in 1X PBS and stored at −80 °C for future use. Total RNA was extracted from the purified exosomes using TRIzol reagent (Thermo Fisher Scientific, Rockford, IL, USA), as per the standard protocol. This RNA served as the basis for subsequent RNA analysis and sequencing.

### 2.4. SmRNA Library Construction and miRNA Sequencing

The smRNA sequencing library was prepared using a TruSeq RNA Sample preparation Kit (Illumina, San Diego, CA, USA), adhering to Illumina’s standard procedure. This preparation involved selecting RNA fractions, ligating adapters, and amplifying the sample to construct the library suitable for high-throughput sequencing. The constructed smRNA library was subsequently sequenced on an Illumina Hiseq 2500 Genome Analyzer platform [26]. The sequencing parameters were set to achieve a read length of 50 base pairs (bp) using single-end sequencing. This approach was chosen to optimize the detection and analysis of smRNA species, particularly miRNAs. The size range of the library was determined to be between 145 bp and 160 bp, which is indicative of successful library preparation and suitable for effective miRNA sequencing. All processes related to the smRNA library construction and sequencing were performed externally by MACROGEN Inc. (Seoul, Republic of Korea), ensuring high-quality and reliable sequencing data.

### 2.5. Data Analysis of miRNA Sequencing

Following smRNA sequencing, the raw sequence data underwent an initial filtering process. This step involved quality-based filtering to separate high-quality reads from the dataset. The processed reads were then further refined by trimming adapter sequences and removing any reads aligned to rRNA sequences. This refinement ensured that the dataset for analysis comprised only relevant and high-quality miRNA sequences. The resulting high-quality, processed reads were then subjected to classification and analysis. Known miRNAs were identified using miRbase v22.1, a comprehensive miRNA sequence database. Other types of RNA sequences were classified using RNAcentral 14.0, a non-coding RNA sequence database. Additionally, the prediction of novel miRNAs was performed using miRDeep2, a tool specifically designed for novel miRNA discovery. The identification of differentially expressed miRNAs was a critical step in this analysis. Statistical methods, including fold change calculation and the exactTest function from the edgeR (version 3.9), were employed. Hierarchical clustering was also utilized to understand the patterns of miRNA expression under different conditions. GO and the KEGG databases were instrumental in analyzing the functions and pathways of the target genes identified in the DE miRNA analysis. These analyses provided insights into the biological implications of the miRNA expression patterns observed. All processes related to the data analysis of miRNA sequencing were conducted by MACROGEN Inc., Seoul, Republic of Korea, ensuring a professional and thorough analysis of the sequencing data. 

### 2.6. Transmission Electron Microscopy (TEM) Analysis

The exosomes extracted from heart tissues were first prepared for TEM analysis by fixing them with a 0.1% paraformaldehyde solution (Biosesang, Seongnam, Republic of Korea) for 30 min. This step is critical in preserving the structural integrity of the exosomes during the subsequent analysis steps. A 10 μL aliquot of each exosome sample was placed on a piece of Parafilm. A formvar/carbon-supported copper grid (200 mesh; Electron Microscopy Sciences, Hatfield, PA, USA) was then floated on top of each sample drop for 7 min [27]. This method allowed the exosomes to adhere to the grid while being sufficiently supported for detailed TEM examination. The grid with adhered exosomes underwent a washing procedure, alternating with three drops of ultrapure water, each wash lasting for a 2 min duration. This step ensured the removal of any residual fixing agent. Subsequently, the grid was stained with a 2% solution of phosphotungstic acid (pH 7.0; Sigma-Aldrich, St. Louis, MO, USA) for 30 s, which provided the contrast necessary to visualize the exosomes under TEM. After staining, the grid was air-dried overnight in a dark environment to prevent any light-induced alterations. Once dried, the exosomes were ready for visualization. The prepared samples were examined under a transmission electron microscope (JEM-F200; JEOL, Tokyo, Japan).

### 2.7. Immunoblot Analysis

Cells were lysed using RIPA buffer (Thermo Fisher Scientific, Rockford, IR, USA), which was supplemented with 1% phosphatase inhibitor (Sigma-Aldrich, St. Louis, MO, USA) and 1% protease inhibitor (Sigma-Aldrich, St. Louis, MO, USA), to ensure effective breakdown of cell components while preserving protein integrity. The protein concentration in the lysates was quantified using the Pierce BCA Protein Assay Kit (Thermo Fisher Scientific, Rockford, IL, USA), enabling accurate determination of protein amounts for uniform loading in gel electrophoresis. The proteins were then separated by electrophoresis on SDS-PAGE under reducing conditions. Following electrophoresis, the proteins were transferred onto a polyvinylidene difluoride (PVDF; Sigma-Aldrich, St. Louis, MO, USA) membrane to prepare for immunoblotting. To prevent non-specific binding, the membrane was blocked for 1 h with 5% skim milk (BD Difco; Sparks, MD, USA) in TBS-T buffer (10 mM Tris-HCl (Sigma-Aldrich, St. Louis, MO, USA), 150 mM NaCl (Sigma-Aldrich, St. Louis, MO, USA), and 0.1% Tween 20 (Sigma-Aldrich, St. Louis, MO, USA)). It was then incubated overnight at 4 °C with primary antibodies (Santa Cruz Biotechnology, Dallas, TX, USA) at a dilution of 1:1000. Following primary antibody incubation, the membrane was washed three times and then incubated with HRP-conjugated anti-mouse IgG (1:1000; Santa Cruz Biotechnology, Dallas, TX, USA) for 2 h in blocking buffer. After three more washes to remove excess secondary antibody, the membrane was prepared for detection [28,29]. Protein bands were visualized using an ECL kit (Western Blotting Detection Kit; GE Healthcare, Buckinghamshire, UK). The band intensities were quantified using ImageJ software (NIH; version 1.54h), providing a quantitative analysis of protein expression.

### 2.8. Isolation of Primary Cardiomyocytes and Fibroblasts from Neonatal Mouse Heart

Primary cardiomyocytes and fibroblasts were isolated from 1-day-old C57BL/6 mice (KOATECH, Pyeongtaek, Republic of Korea) using the Primary Cardiomyocyte Isolation Kit (Thermo Fisher Scientific, Rockford, IL, USA) and Primary Fibroblast Isolation Kit (Thermo Fisher Scientific, Rockford, IL, USA), respectively, following the manufacturer’s protocols. Neonatal hearts were dissected into 1–3 mm^3^ pieces and initially placed separately in chilled Hank’s Balanced Salt Solution (HBSS; Thermo Fisher Scientific, Grand Island, NY, USA). After two washes with 0.5 mL of cold HBSS, the heart pieces were subjected to enzymatic digestion. For primary cardiomyocyte isolation, each heart in a tube was treated with 0.5 mL of Enzyme 1 (containing papain) and 0.01 mL of Cardiomyocyte Isolation Enzyme 2 (containing thermolysin), and incubated at 37 °C for 30 min. For primary fibroblast isolation, each heart was treated with 0.2 mL of reconstituted MEF Isolation Enzyme (containing papain) and incubated at 37 °C for 25 min. Following enzymatic digestion, the heart tissues were washed twice with 0.5 mL of cold HBSS. The tissues were then mechanically disrupted by pipetting up and down 25 times for primary cardiomyocytes or 20 times for primary fibroblasts in 0.5 mL of complete DMEM (Thermo Fisher Scientific, Grand Island, NY, USA) containing 10% fetal bovine serum (FBS; HyClone, Logan, UT, USA) and 1% penicillin/streptomycin (Thermo Fisher Scientific, Grand Island, NY, USA). The resultant cell suspensions were combined, and cell concentration and viability were determined. The cells were then seeded at a density of 4 × 10^4^ cells/well in a 96-well plate (SPL, Pocheon, Republic of Korea) for transfection with miRNA mimics. For immunofluorescence analysis, cells were seeded at 2 × 10^5^ cells/well in a 4-well cell culture slide (SPL, Pocheon, Republic of Korea).

### 2.9. Transfection with miRNA Mimics

Primary cardiomyocytes and fibroblasts were seeded in 96-well plates in preparation for transfection. On the following day, each well was transfected with 20 different miRNA mimics at a concentration of 1 pmol/well using Lipofectamine RNAiMAX (Thermo Fisher Scientific, Rockford, IL, USA) according to the manufacturer’s guidelines. A miRNA mimic negative control was utilized for comparison [30]. All miRNA mimics used in this study were sourced from Genolution Pharmaceuticals (Seoul, Republic of Korea), with their specific details provided in Table 1. For control 24 h post-transfection, the cells were shifted to serum-free medium (SFM) and subjected to either normoxic or hypoxic conditions for an additional 24 h. This setup was designed to simulate an in vitro environment analogous to MI. For hypoxic treatment, cells were maintained at 37 °C in 5% CO_2_, 5% H_2_, and 0.5% O_2_, facilitated by an anaerobic atmosphere system (Technomart, Seoul, Republic of Korea).

### 2.10. Cytotoxicity Assay

In evaluating cytotoxicity in the cell cultures, the ToxiLight BioAssay Kit (Lonza, Walkersville, MD, USA) was employed. This kit is a non-destructive cytolysis assay, specifically designed to measure the release of adenylate kinase (AK) from damaged cells. The assay utilizes bioluminescent reaction that correlates with the amount of AK released from lysed cells. For the assay, 0.02 mL of cell culture supernatant was transferred to a new 96-well plate. To this, 0.1 mL of AK test reagent, dissolved in assay buffer, was added. This mixture was then incubated at room temperature for 5 min to allow for the development of the bioluminescent reaction [31,32]. Following the incubation, the bioluminescence intensity was measured using a GloMax Discover Microplate Reader (Promega, Madison, WI, USA).

### 2.11. Immunofluorescence Analysis

The cell culture slides containing primary cardiomyocytes and fibroblasts were fixed in 4% paraformaldehyde solution (Biosesang, Seongnam, Republic of Korea) overnight at 4 °C. Following fixation, antigen retrieval was performed using a sodium citrate buffer (0.1 M; CureBio, Seoul, Republic of Korea) for 10 minutes at 95 °C. The slides were then permeabilized in 0.2% Triton X-100 (Sigma-Aldrich, St. Louis, MO, USA) for 10 minutes, allowing antibody access to intracellular structures. Subsequently, the slides were blocked with 2.5% normal horse serum (Vector Laboratories, Newark, CA, USA) for 1 h to minimize non-specific binding of the antibodies. After blocking, the slides were incubated with primary antibodies: either anti-cardiac troponin T antibodies (1:200 dilution; Abcam, Cambridge, UK) for cardiomyocytes or anti-vimentin antibodies (1:200 dilution; Abcam, Cambridge, UK) for fibroblasts, overnight at 4 °C. Post-primary antibody incubation, the slides were washed and incubated with appropriate secondary antibodies: either fluorescein isothiocyanate (FITC)-conjugated secondary antibodies (1:500 dilution; Jackson Immunoresearch, West Grove, PA, USA) or rhodamin-conjugated secondary antibodies (1:500 dilution; Millipore, Bedford, MA, USA). Nuclei were stained with 4′,6-diamidino-2-phenylindole (DAPI; 1:5000 dilution; Thermo Fisher Scientific, Rockford, IL, USA) to facilitate the identification of cell structures [30,33]. Finally, the slides were examined under an Olympus IX83 microscope (Evident, Tokyo, Japan) for detailed visualization of the immunofluorescence staining.

### 2.12. Statistical Analysis

The data were analyzed through two-sample t-test using Statistical Package for the Social Sciences (SPSS, version 14.0K) software and results are expressed as the means ± standard error of the means (SEMs). When the t-test indicated a significant overall treatment effect (*p* < 0.05), differences between groups were assessed using the least-significant difference (LSD) test, with significance set at *p* < 0.05. The *p* value from the RNA sequencing analysis was automatically extracted by a comparative analysis algorithm called edgeR.

## 3. Results

### 3.1. Establishment of MI Animal Models and Isolation of Exosomes from Heart Tissues

ExoRNA sequencing was conducted on exosomes extracted from heart tissues of an established MI mouse model using C57BL/6 mice (Figure 1A). Five mice per group were used for sequencing analysis. To verify MI induction in these models, the heart morphology of the MI models was compared with that of the sham group. Infarction sites were evident in the heart of MI models (Figure 1B). Additionally, the infarcted areas of hearts were compared using TTC staining. This is because TTC staining reacts with mitochondrial enzymes in living cells to form a red compound. Uniform staining was observed in the sham group hearts, while the MI groups exhibited TTC-negative areas at the infarction site (Figure 1B). Exosome size extracted from heart tissues was confirmed using TEM analysis (Figure 1C). Immunoblotting was employed to verify the presence of exosome markers including CD9, CD64, and CD81, with β-actin, α-tubulin, and GAPDH serving as internal controls (Figure 1D).

### 3.2. ExoRNA Sequencing and Data Processing

ExoRNA sequencing was conducted on exosomes extracted from the heart tissues of C57BL/6 mice in an MI model. Sequencing adapters were ligated to the exoRNA from heart tissue, followed by reverse transcription (RT) and PCR to amplify cDNA pools (Figure 2A). These cDNA fragments underwent sequencing on an Illumina platform. Data processing involved organizing sequenced reads into categories: trimmed reads (with adapter sequences removed), nonadapter reads (without adapter sequences), short reads (less than 17 bp post-adapter trimming), and low-quality reads (with one or more bases in the trimmed or non-adapter reads). Read counts of each group are presented in Figure 2A. Ribosomal RNA (rRNA) removal was performed to mitigate its abundance effect, with the remaining read counts shown in Figure 2B. Read length distribution across each group is depicted in Figure 2C. Generally, transfer RNA (tRNA) was 70–90 nucleotide (nt), small nucleolar RNA (snoRNA) was about 90 nt, small nuclear RNA (snRNA) was between 100 and 300 nt, miRNA was about 22 nt and PIWI-interacting RNA (piRNA) was about 27 nt in length. The small RNA (smRNA) composition of each sample, indicating the ratio of smRNA types (such as known miRNA, candidate miRNA, rRNA, tRNA, snRNA, snoRNA, etc.), is illustrated in Figure 2D.

### 3.3. Differential Expressed (DE) miRNA Analysis 

To assess the similarity between samples, correlation analysis and hierarchical clustering analysis were performed. In the correlation analysis, a value closer to 1 indicates greater similarity. The highest similarity was observed between the sham group and the MI-1day group, followed by the MI-1day and MI-3day groups. The lowest similarity was noted between the sham group and the MI-3day group (Figure 3A). Hierarchical clustering analysis further supported these findings, showing a high similarity between the sham and MI-1day groups and a lower similarity between the combined sham/MI-1day group and the MI-3day group (Figure 3B, left). A heat map, utilizing the Euclidean method and complete linkage in the hierarchical clustering analysis, clustered the mature miRNAs and samples based on their expression levels (normalized value). This clustering highlighted significant differences between at least one pair of total comparison groups (Figure 3B, right). A total of 174 mature miRNAs, which met the criteria of a fold change (|FC| > 2) and *p*-value (*p* < 0.05), were categorized according to differential expression between groups (Figure 3C). These findings were visually represented in smear plots of the average logCPM (*X*-axis) and log2 fold change (*Y*-axis), to examine transcripts showing notable expression differences (Figure 3D).

To elucidate the function of exoRNAs from heart tissues in the MI animal model and identify enriched functional terms, gene ontology (GO) and Kyoto Encyclopedia of Genes and Genomes (KEGG) pathway analyses were conducted on the differentially expressed mRNAs of dysregulated miRNAs. These analyses were performed separately for miRNAs, showing increased and decreased expressions between MI-1day and sham/MI-3day and sham groups (Figure 4). The GO analysis encompassed three categories: biological process, cellular component, and molecular function. The top seven results from the GO enrichment analysis are presented in Figure 4A. In the MI-1day group, the most enriched biological process terms included organelle organization, regulation of molecular function, and nervous system development. Conversely, in MI-3day group, the terms focused on cell development, protein localization, and cellular catabolic process. The primary cellular component terms for the MI-1day group were cell projection, neuron projection, and vesicle, while for MI-3day group, they were neuron, golgi apparatus, and endomembrane system. The molecular function terms common to both MI-1day and MI-3day groups were ion binding, cation binding, and transition metal ion binding. These findings suggest the involvement of certain miRNAs in the fundamental biological regulation of MI. Additionally, the KEGG pathway analysis highlighted key pathways showing significant differences between MI and sham groups, including endocytosis, the mitogen-activated protein kinase (*MAPK*) signaling pathway, cyclic adenosine monophosphate (*cAMP*) signaling pathway, phosphoinositide 3-kinase (*PI3K*)-*Akt* signaling pathway, and *Ras* signaling pathway (Figure 4B).

### 3.4. Selection of Differentially Expressed miRNAs 

To determine which miRNAs to investigate in vitro from those differentially expressed miRNAs among the groups, a diagram was initially used to chart both increased and decreased miRNAs (Figure 5A). The primary focus of this study was on miRNAs that showed reduced expression in the MI group, as this approach allowed for covering a broader range of miRNAs in the in vitro studies. Subsequently, miRNAs already known to be associated with MI and those exhibiting very low expression levels were excluded to refine the selection of candidate miRNAs for in vitro analysis. Consequently, out of the 49 miRNAs that demonstrated decreased expression in MI groups, 20 were selected for the in vitro functional study (Figure 5B).

### 3.5. Effects of Selected miRNAs on Hypoxic Stress-Induced Cell Death

To simulate MI conditions in vitro, primary cardiomyocytes and fibroblasts were isolated from neonatal mouse hearts and subjected to hypoxic stress. These cell types were chosen because they are the most abundant in heart tissue. Following isolation, the cells were characterized using specific cell markers: troponin-T for cardiomyocytes, and vimentin for fibroblasts (Figure 6, left). The cells were then transfected with mimics of 20 selected miRNAs and exposed under hypoxic condition to assess the effects on cell viability. These observations revealed that 19 of the selected miRNAs, with the exception of miR-1247-5p, significantly reduced hypoxic stress-induced cell death in primary cardiomyocytes. In primary fibroblasts, 12 miRNAs (miR-30c-1-3p, miR-149-3p, miR-206-3p, miR-486a-3p, miR-673-3p, miR-690-3p, miR-700-5p, miR-706, miR-744-5p, miR-871-3p, miR-874-5p, and miR-1247-5p) effectively attenuated cytotoxicity under low oxygen conditions (Figure 6, right).

## 4. Discussion

### 4.1. Establishment of MI Model

This study provided a comprehensive profile of exoRNAs, particularly miRNAs, in an MI mouse model established using RNA sequencing techniques. Additionally, the effects of differentially expressed miRNAs in an MI group on hypoxic-induced cell death were also examined in primary cardiomyocytes and fibroblasts. A key achievement of this study was the successfully establishment of an MI animal model using C57BL/6 mice. The occurrence of left ventricular dysfunction and heart failure in rats is an acute renal muscle infarction model, and is primarily used to study the function of MI [34]. Mouse models of MI based on C57BL/6 mice have also been widely used in the study of MI and heart failure [35,36]. Morphological changes in the MI model, confirmed by TTC staining, clearly distinguished between infarcted and non-infarcted areas, validating the model’s efficacy (Figure 1B). MI is typically characterized by three phases: the inflammatory phase, proliferative phase, and the maturation phase [37]. During the inflammatory phase, cardiomyocyte death, proinflammatory cytokines secretion, and neutrophil infiltration occur. The proliferative phase is marked by macrophage polarization, myofibroblast proliferation, and collagen deposition. Finally, the maturation phase involves extracellular matrix cross-linking, myofibroblast quiescence, and heart failure. In the present study, the MI mouse model during the acute inflammatory phase of MI was utilized to identify potential biomarkers that could be useful for early-stage MI detection and intervention. 

### 4.2. Exosome Analysis 

Furthermore, exosomes were isolated from cardiac tissues and their identity through TEM and immunoblotting analyses was confirmed (Figure 1C,D). Exosomes are heterogeneous populations of 50–250 nm membrane-bound vesicles that contain proteins, lipids, and nucleic acids, playing crucial roles in various disease processes [38]. While exosomes are abundantly present in blood vessels, they are also found in the interstitial spaces of cellular tissues [39]. The exosome analyzed in this study were specifically derived from cells within the heart tissues affected by MI, focusing on those newly released by cells post-MI, rather than those of unclear origin circulating in the blood. 

### 4.3. RNA Sequencing and Analysis 

In this study, comprehensive RNA sequencing and data processing of exoRNAs from MI models revealed a diverse spectrum of RNA species. This spectrum included microRNAs (miRNAs), transfer RNAs (tRNAs), small nuclear RNAs (snRNAs), small nucleolar RNAs (snoRNAs), and PIWI-interacting RNAs (piRNAs). The analysis of read length distribution and smRNA composition in this study offers an extensive overview of the exoRNA profile in MI (Figure 2). Such detailed information is vital for deciphering the molecular mechanisms at play in MI, suggesting these RNAs’ potential roles in intercellular communication and the cardiac injury response. The DE analysis of miRNAs underlines significant changes in miRNA expression in response to MI (Figure 3). Correlation and hierarchical clustering analyses (Figure 3) indicate that the longer the ligation duration of the left anterior descending (LAD) artery, the more pronounced the differences in miRNA expression between the sham and MI groups. These insights are pivotal for understanding the time-dependent regulation of gene expression in the context of MI. Furthermore, they could be instrumental in guiding the development of targeted therapeutic interventions.

The functional enrichment analysis, utilizing both GO and KEGG pathways, shed light on the biological processes, cellular components, and molecular functions altered in MI (Figure 4). Particularly significant is the identification of pathways such as *MAPK*, *cAMP*, *PI3K-Akt*, and *RAS* signaling pathways in relation to MI-associated mRNAs. Notably, the *PI3K*/*Akt* pathway is known to regulate the growth and survival of cardiomyocytes and plays a critical role in the pathophysiology of MI [40]. Similarly, the *cAMP* signaling pathway and its compartmentalization are pivotal in cardiac physiology [41]. The *MAPK* is involved in the proliferation of cardiac fibroblast and interaction between fibroblasts and cardiomyocytes [42], while *RAS* signaling stimulates cardiomyocyte hypertrophy and fibroblast proliferation [43]. Thus, the functional enrichment analysis (Figure 4) corroborates the findings of numerous previous studies. These pathways are implicated in various aspects of cardiac function and pathology, including cell survival, proliferation, and apoptosis. This underscores the potential of identified miRNAs as therapeutic targets or biomarkers in the context of cardiac health and disease.

### 4.4. Functional Assay

The meticulous selection of miRNAs for in vitro functional studies, based on their differential expression and prior associations with MI, represents a systematic approach to uncovering novel factors in MI pathology (Figure 5). By deliberately excluding miRNAs that are already well-established roles in MI research, or those exhibiting low expression levels, this study concentrated on potentially novel miRNAs. This strategy aimed at revealing new insights into the mechanisms of MI. Consequently, a set of 20 miRNAs were identified as promising candidates for further investigation in the in vitro functional study, potentially acting as causative factors in MI.

Primary cardiac cells or cardiac cell lines exposed to hypoxic stress are often used as in vitro models of myocardial infarction [44,45,46], because AMI causes hypoxic stress in cardiac cells [47]. In this study, primary cardiomyocytes and fibroblasts as in vitro models of myocardial infarction were used. The mammalian adult heart is composed of various cell types, including cardiomyocyte, fibroblast, endothelial cells, and peri-vascular cells [48]. Studies by Bannerjee et al. reported that the rat heart comprises 30% cardiomyocyte, 64% fibroblast, and 6% endothelial cells. In contrast, the mouse heart consists of 54% cardiomyocyte, 26% fibroblast, and 6% endothelial cells [49]. Based on this cellular composition, this study primarily utilized cardiomyocyte and fibroblasts to investigate the functional role of miRNAs. The in vitro experiments, conducted under hypoxic conditions on primary cardiomyocytes and fibroblasts, simulated the MI environment, thereby providing insights into the protective roles of selected miRNAs (Figure 6).

Figure 6 illustrates that 19 and 12 miRNAs significantly inhibited hypoxic stress-induced cell death in primary cardiomyocytes and fibroblasts, respectively. Notably, 11 miRNAs demonstrated protective effects against hypoxic cell death in both cell types. Interestingly, most of the 20 miRNAs selected had no previously reported association with the heart. However, there are a few studies on some of these miRNAs in different contexts. For example, miR-30c-1-3p was significantly downregulated in retinas of mice with oxygen-induced retinopathy [50], and miR-149 was shown to enhance the myocardial differentiation of mouse bone marrow stem cells [51]. Additionally, miR-706 was reported to block oxidative stress-induced activation of liver in fibrogenesis [52]. The observation that most of the selected miRNAs significantly reduced hypoxic stress-induced cell death in both cardiomyocytes and fibroblasts (Figure 6) is a promising indication of their potential therapeutic utility in treating MI. 

### 4.5. Limitations, Strength, and Perspectives of This Study 

The limitations of this study include its reliance on in vitro and animal models, which may not fully replicate in vivo the human condition of AMI. Additionally, while miRNAs that can inhibit hypoxia-induced cell death were identified in this study, limitations to the sensitivity and specificity of a single miRNA still exist. Further research is required to validate these findings in human subjects and explore the therapeutic potential of these miRNAs in clinical settings. 

On the other hand, a notable aspect of this study is its focus on miRNAs showing reduced expression in the MI group. This approach facilitated the exploration of potentially novel miRNAs in the context of MI, extending beyond those already well-characterized in this disease. The in vitro functional studies revealed that selected miRNAs significantly mitigated hypoxic stress-induced cell death in primary cardiomyocytes and fibroblasts, suggesting their protective roles in the context of MI and highlighting their potential as therapeutic targets or biomarkers. Furthermore, the strengths of this study lie in derivation of miRNAs from exosomes secreted from heart tissue directly affected by MI. As these miRNAs closely reflect the heart’s condition during MI, the study’s results realistically portray the characteristics of MI-secreted exosomes. There is a similar report about plasma exosomal miRNAs in human AMI samples, although the origin of species and tissues differ [53]. This report, which examined miRNA profiles from the plasma of 118 subjects and identified 18 miRNAs as biomarkers for early AMI diagnosis, highlights both the advantages and disadvantages of the current study.

This discovery not only broadens understanding of miRNA-mediated regulation in cardiac tissues but also emphasizes their potential as therapeutic targets for AMI. Future research should aim at the clinical translation of these findings, striving to improve the diagnosis, treatment, and prognosis of AMI.

## 5. Conclusions

This study offers a comprehensive analysis of exoRNAs within an MI model, underlining the dynamic alterations in miRNA expression and exploring their potential functional roles. By integrating RNA sequencing data with functional in vitro assays, a robust framework for elucidating the molecular mechanisms underlying MI has been established, pinpointing novel therapeutic targets. Looking ahead, it is imperative to extend this research by validating these findings in human samples. Doing so would not only reinforce the relevance of these results but also bridge the gap between experimental models and clinical applications. Furthermore, investigating the therapeutic potential of these miRNAs in clinical settings stands as a crucial next step. This would involve evaluating their efficacy and safety in human subjects, potentially opening new avenues for MI treatment and management.

## Figures and Tables

**Figure 1 biomedicines-12-00430-f001:**
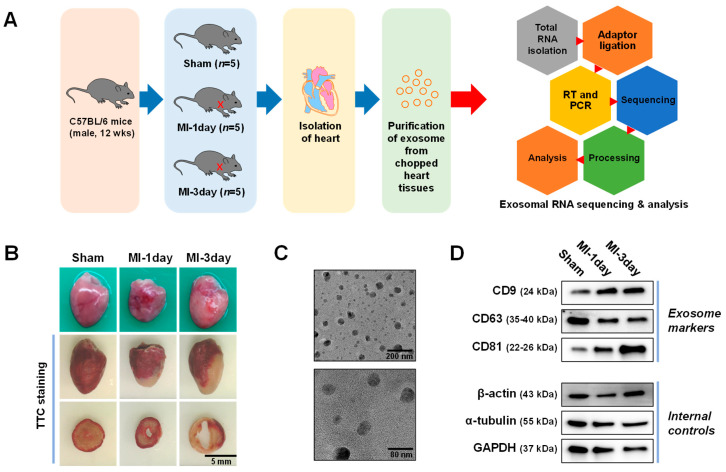
Experimental overview and exosome characterization of extracted from heart tissues. (**A**) Schematic representation of the experimental procedure employed in the current study. (**B**) Group-specific heart images from the myocardial infarction (MI) mouse models, both with and without triphenyltetrazolium chloride (TTC) staining, illustrating the morphological differences among the groups. (**C**) Transmission electron microscopy (TEM) images of isolated exosome stained with phosphotungstic acid, showcasing their structural features. (**D**) Immunoblotting analysis of the isolated exosomes using exosomal markers, including CD9, CD63, and CD81, alongside internal controls. Group designations: sham, control group; MI-1day, MI induced for 1 day; MI-3day, MI induced for 3 days.

**Figure 2 biomedicines-12-00430-f002:**
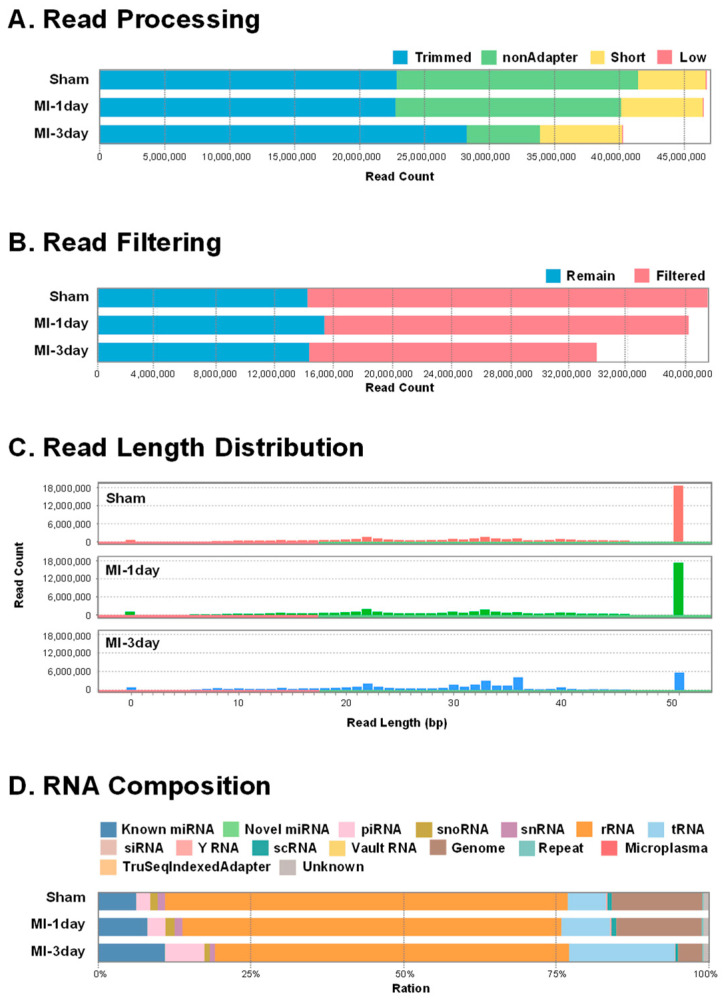
Comprehensive analysis of exosomal RNA (exoRNA) sequencing data. (**A**) Depicts the count distribution of various read types for each group, including trimmed reads, nonadapter reads, short reads and low-quality reads. (**B**) Shows the remaining reads (blue) after the removal of ribosomal RNA (rRNA; red). (**C**) Illustrates the distribution of read lengths across each sample. (**D**) Provides a breakdown of the smRNA composition within each sample, categorizing them into various types: miRNA (microRNA), piRNA (PIWI-interacting RNA), snoRNA (small nucleolar RNA), snRNA (small nuclear RNA), rRNA (ribosomal RNA), tRNA (transfer RNA), siRNA (small interfering RNA), Y RNA, scRNA (single-cell RNA).

**Figure 3 biomedicines-12-00430-f003:**
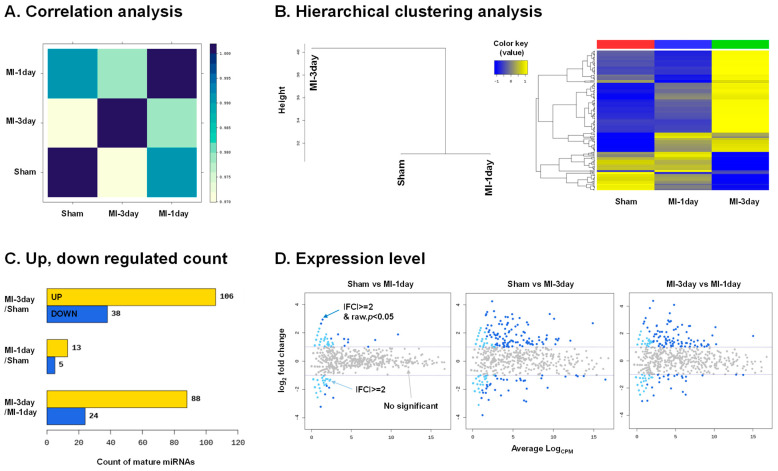
Differential Expression (DE) miRNA analysis from exosomal RNA (exoRNA) sequencing. (**A**) Correlation matrix: This panel presents the correlation matrix of all samples, calculated using Pearson’s coefficient based on normalized values. The correlation coefficient (r) ranges from −1 to 1, where values closer to 1 indicate a higher similarity between samples. (**B**) Hierarchical clustering: The left part of this panel shows the hierarchical clustering of samples based on their normalized expression normalized value, where samples with higher expression similarities are grouped together (distance metric = Euclidean distance, linkage method = complete linkage) (left). The right part features a heat map of the two-way hierarchical clustering, utilizing Z-scores of Log2-transformed normalized values for visualization. (**C**) Quantitative analysis of microRNAs (miRNAs): This section displays the number of mature miRNAs that are either up-regulated or down-regulated based on a fold change (|FC| > 2) and *p*-value (*p* < 0.05) for each comparison pair. (**D**) Smear plots: This panel illustrates smear plots representing the expression level of miRNAs. The plots are designed to visually represent the distribution and variance of miRNA expression across samples.

**Figure 4 biomedicines-12-00430-f004:**
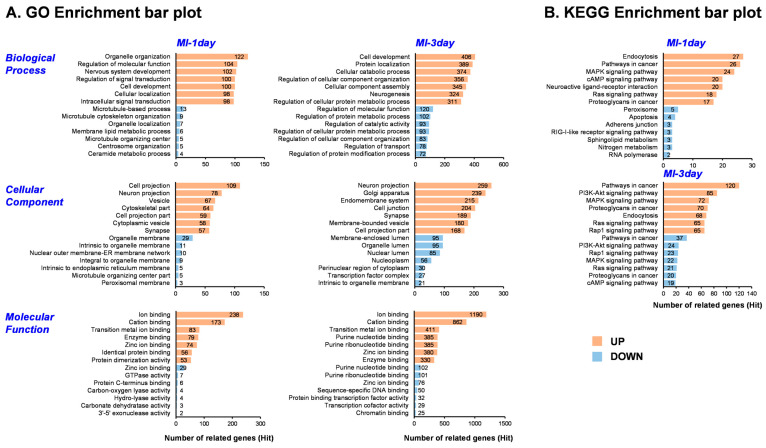
Comprehensive functional enrichment analysis of differentially expressed miRNAs. (**A**) Depicts the top seven gene ontology (GO) terms, providing insights into the biological processes, cellular components, and molecular functions most affected by the dysregulated miRNAs. (**B**) Illustrates the top seven pathways identified in the Kyoto Encyclopedia of Genes and Genomes (KEGG) pathway analysis, highlighting critical pathways impacted by the altered expression of mRNAs across different groups.

**Figure 5 biomedicines-12-00430-f005:**
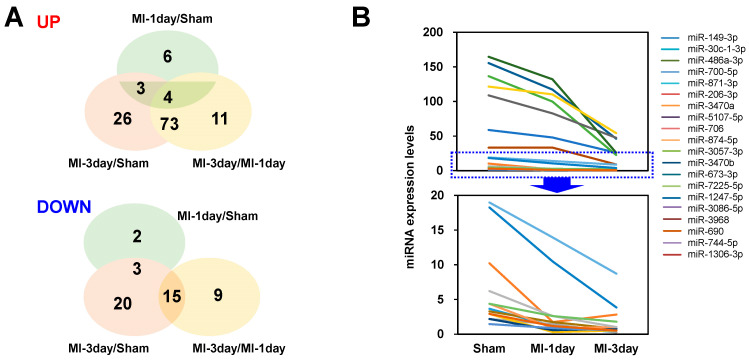
Identification and selection of key candidate microRNAs (miRNAs). (**A**) A diagrammatic representation of the number of miRNAs exhibiting increased (up) or decreased (down) expression upon comparative analysis between groups. This visualization aids in understanding the overall distribution and directional trends of miRNA expression changes. (**B**) A broken line graph displaying the expression profiles of the 20 selectively chosen miRNAs across the groups, enabling a comparative and detailed view of their expression dynamics.

**Figure 6 biomedicines-12-00430-f006:**
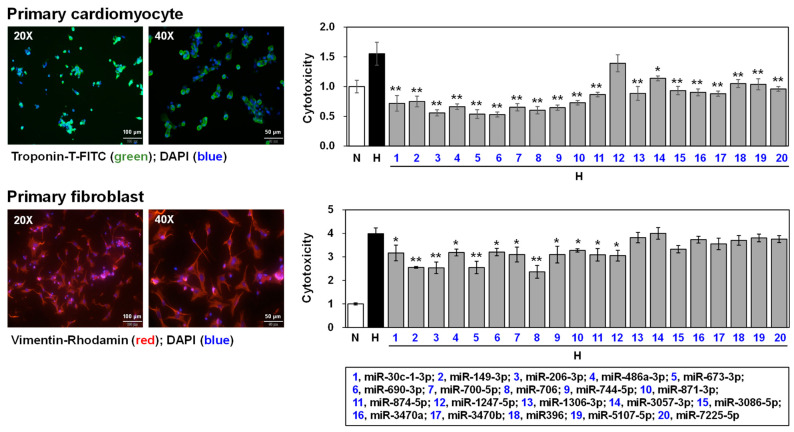
Analysis of cytotoxicity using selected miRNA mimics in primary cardiomyocytes and fibroblasts. On the left, immunofluorescence analysis confirms the presence of specific markers in isolated cells: troponin-T in primary cardiomyocytes and vimentin in primary fibroblasts. The right side of the figure investigates the impact of miRNA mimic treatment on cell survival under hypoxic conditions. In the accompanying bar graph, white and black bars represent the control (miRNA mimic negative control-treated) group, while gray bars denote groups treated with miRNA mimics. Statistical significance between the control and treated groups was assessed using ANOVA. *p* value annotations indicate levels of significance: * *p* < 0.05 and ** *p* < 0.01, highlight key differences in cytotoxicity responses. The error bar represents the standard deviation between the five wells of the 96-well plates used in the cytotoxicity assay. N, normoxic condition; H, hypoxic condition; FITC, fluorescein isothiocyanate; DAPI, 4′,6-diamidino-2-phenylindole; miR, microRNA.

**Table 1 biomedicines-12-00430-t001:** Information about miRNA mimics used for transfection assay.

No.	Target Name	Accession#	Sequence (5′–3′)
1	mmu-miR-30c-1-3p	MIMAT0004616	CUGGGAGAGGGUUGUUUACUCC
2	mmu-miR-149-3p	MIMAT0016990	GAGGGAGGGACGGGGGCGGUGC
3	mmu-miR-206-3p	MIMAT0000239	UGGAAUGUAAGGAAGUGUGUGG
4	mmu-miR-486a-3p	MIMAT0017206	CGGGGCAGCUCAGUACAGGAU
5	mmu-miR-673-3p	MIMAT0004824	UCCGGGGCUGAGUUCUGUGCACC
6	mmu-miR-690	MIMAT0003469	AAAGGCUAGGCUCACAACCAAA
7	mmu-miR-700-5p	MIMAT0017256	UAAGGCUCCUUCCUGUGCUUGC
8	mmu-miR-706	MIMAT0003496	AGAGAAACCCUGUCUCAAAAAA
9	mmu-miR-744-5p	MIMAT0004187	UGCGGGGCUAGGGCUAACAGCA
10	mmu-miR-871-3p	MIMAT0017265	UGACUGGCACCAUUCUGGAUAAU
11	mmu-miR-874-5p	MIMAT0017268	CGGCCCCACGCACCAGGGUAAG
12	mmu-miR-1247-5p	MIMAT0014800	ACCCGUCCCGUUCGUCCCCGGA
13	mmu-miR-1306-3p	MIMAT0009411	ACGUUGGCUCUGGUGGUGAUG
14	mmu-miR-3057-3p	MIMAT0014823	UCCCACAGGCCCAGCUCAUAGC
15	mmu-miR-3086-5p	MIMAT0014880	UAGAUUGUAGGCCCAUUGGA
16	mmu-miR-3470a	MIMAT0015640	UCACUUUGUAGACCAGGCUGG
17	mmu-miR-3470b	MIMAT0015641	UCACUCUGUAGACCAGGCUGG
18	mmu-miR-3968	MIMAT0019352	CGAAUCCCACUCCAGACACCA
19	mmu-miR-5107-5p	MIMAT0020615	UGGGCAGAGGAGGCAGGGACA
20	mmu-miR-7225-5p	MIMAT0028418	ACGUAGACUGUGUAGAAGCC

## Data Availability

The data that support the findings of this study are available on request from the corresponding author.

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
