# Peer review of "Exploring Cardiac Exosomal RNAs of Acute Myocardial Infarction"

_biomedicines, 2024, doi:10.3390/biomedicines12020430_

Round 1
Reviewer 1 Report
Comments and Suggestions for Authors
Congratulations. Excellent study. However, the presentation must be improved. Please, see in the attached file, my comments and suggestions.

Author Response
Responses to the reviewer’s comments (Biomedicines-2844574)
We would like to thank the reviewer again for their careful reading and valuable comments that have really helped us improve the current revised version of our manuscript. We have addressed all the general and specific comments made by the referee in order to improve the clarity of the manuscript. We agree with most of his/her propositions and updated our manuscript accordingly. Moreover, and in order to help the assessors in their reviewing process, we have marked all the new changes and corrections in the revised version in blue. A full point-by-point list of responses and explanations is provided below.
C: comments by the reviewer; R: responses by the authors
Reviewer #1:
Congratulations. Excellent study. However, the presentation must be improved. Please, see in the attached file, my comments and suggestions.
[C1] I suggest trying to avoid the personal format (we, our) in the manuscript, here and throughout the manuscript.
[R1] In response to the reviewer’s insightful suggestions, the personal format, including terms like 'we' and 'our,' has been eliminated throughout the revised manuscript.
[C2] Change to “among”
[R2] The term 'between' has been replaced with 'among' in the revised manuscript, as suggested. This change can be found on line 47.
[C3] Add reference about the method.
[R3] As recommended, references detailing the method for establishing the MI mouse model have been added to the revised manuscript. These can be found in the references section as numbers 21 and 22.
[C4] Add reference about the method.
[R4] In accordance with the suggestion, a reference detailing the method for TTC staining has been included in the revised manuscript. This reference is listed as numbers 23 and 24.
[C5] Add information about the method used. Please clarify.
[R5] Additional information about the TTC staining method has been incorporated into the revised manuscript for clarity.
[C6] Add reference about the method.
[R6] As recommended, references detailing the method for exosome isolation have been added to the revised manuscript. These can be found in the references section as number 25.
[C7] Add reference about the method.
[R7] In response to the request, a reference detailing the method for SmRNA library construction and miRNA sequencing has been added to the revised manuscript, cited as reference number 26.
[C8] Add reference about the method.
[R8] A reference describing the method used for TEM analysis has been added to the revised manuscript, listed as reference number 27.
[C9] Add reference about the method.
[R9] In compliance with the suggestion, a reference detailing the method for immunoblot analysis has been included in the revised manuscript. This reference is cited as numbers 28 and 29.
[C10] Add reference about the method.
[R10] As requested, a reference detailing the method used for transfection with miRNA mimics has been added to the revised manuscript. This can be found listed as reference number 30.
[C11] Add reference about the method.
[R11] In accordance with the suggestion, a reference describing the method for the cytotoxicity assay has been included in the revised manuscript. This reference is cited as numbers 31 and 32.
[C12] Add reference about the method.
[R12] A reference detailing the method used for immunofluorescence analysis has been added to the revised manuscript, listed as reference numbers 30 and 33.
[C13] I suggest adding 3 paragraphs at the end of the Discussion section:
1- with the limitations of the study
2- with the strengths of the study
3- with facts and perspectives or future directions
[R13] In response to the reviewer's recommendation, three new paragraphs have been added to the end of the Discussion section in the revised manuscript. These paragraphs address the limitations, strengths of the study, and provide facts and perspectives for future directions, specifically found in lines 517-541.
Reviewer 2 Report
Comments and Suggestions for Authors
Jung et al reported their work named "Exploring cardiac exosomal RNAs of acute myocardial infarction" and concluded "Our research enhances understanding of post-AMI molecular changes in cardiac tissues and investigates the potential of exoRNAs as biomarkers or therapeutic targets. These findings offer new insights into AMI's molecular mechanisms, paving the way for RNA-based diagnostics and therapeutics and therapies and contributing to the advancement of cardiovascular medicine.". I have the following comments:
- Please edit this sentence in the abstract "MI), often the first symptom of coronary artery" to be "MI), a frequent symptom of coronary artery..".
- Please specify the number of mice in the abstract results and in the manuscript text.
- Please spell out abbreviations at their first time eg "miRNAs" , "PBS", "HEPES", "TTC', SEM and 'FITC", and "DAPI" in figure 6
- Language revision is essential eg "sorbin And SH3.." should be "sorbin and SH3..", "mice were airway ventilation using..'
- Statistical analysis: I believe you mean two-way ANOVA rather than one-way ANOVA. Please edit that, otherwise, please explain the rationale with reference.
- Figure 5B: Please separate the Venn diagram circles if no intersection i.e. the yellow and green ones.
- Figure 6: please specify the meaning of error bars eg SD or 95% confidence intervals in the figure legend
- This is a similar work that should be discussed "Link: https://www.ncbi.nlm.nih.gov/pmc/articles/PMC7940945/"
Comments on the Quality of English LanguagePlease see above
Author Response
Responses to the reviewer’s comments (Biomedicines-2844574)
We would like to thank the reviewer again for their careful reading and valuable comments that have really helped us improve the current revised version of our manuscript. We have addressed all the general and specific comments made by the referee in order to improve the clarity of the manuscript. We agree with most of his/her propositions and updated our manuscript accordingly. Moreover, and in order to help the assessors in their reviewing process, we have marked all the new changes and corrections in the revised version in blue. A full point-by-point list of responses and explanations is provided below.
C: comments by the reviewer; R: responses by the authors
Reviewer #2:
Jung et al reported their work named "Exploring cardiac exosomal RNAs of acute myocardial infarction" and concluded "Our research enhances understanding of post-AMI molecular changes in cardiac tissues and investigates the potential of exoRNAs as biomarkers or therapeutic targets. These findings offer new insights into AMI's molecular mechanisms, paving the way for RNA-based diagnostics and therapeutics and therapies and contributing to the advancement of cardiovascular medicine.". I have the following comments:
[C1] Please edit this sentence in the abstract "MI), often the first symptom of coronary artery" to be "MI), a frequent symptom of coronary artery..".
[R1] The word 'first' has been replaced with 'frequent' in the Abstract of the revised manuscript, specifically in line 13.
[C2] Please specify the number of mice in the abstract results and in the manuscript text.
[R2] In accordance with the reviewer's insightful comment, the number of mice used in each group has now been specified in both the abstract results and within the manuscript text of the revised manuscript, specifically on lines 23 and 273.
[C3] Please spell out abbreviations at their first time eg "miRNAs" , "PBS", "HEPES", "TTC', SEM and 'FITC", and "DAPI" in figure 6
[R3] All abbreviations, including 'miRNAs', 'PBS', 'HEPES', 'TTC', 'SEM', 'FITC', and 'DAPI', have been spelled out at their first mention in both the text and the legend of Figure 6 in the revised manuscript.
[C4] Language revision is essential eg "sorbin And SH3.." should be "sorbin and SH3..", "mice were airway ventilation using..'
[R4] The necessary language revisions have been made in the revised manuscript. 'And' has been changed to 'and' and 'ventilation' has been corrected to 'ventilated.' These changes can be found in lines 61 and 89, respectively.
[C5] Statistical analysis: I believe you mean two-way ANOVA rather than one-way ANOVA. Please edit that, otherwise, please explain the rationale with reference.
[R5] We sincerely apologize for our significant error. The data in Figure 6 were analyzed using a two-sample t-test through SPSS software. Additionally, the p-value from the RNA sequencing analysis is derived automatically by a comparative analysis algorithm known as edgeR. These details have been added to the Materials and Methods section of the revised manuscript, specifically between lines 264-269.
[C6] Figure 5B: Please separate the Venn diagram circles if no intersection i.e. the yellow and green ones.
[R6] In response to the reviewer’s comment, the circles in the Venn diagram that do not intersect, specifically the yellow and green ones, have been separated in Figure 5 of the revised manuscript.
[C7] Figure 6: please specify the meaning of error bars eg SD or 95% confidence intervals in the figure legend
[R7] Following the reviewer's suggestion, we have clarified in the legend of Figure 6 that the error bars represent the standard deviation calculated from the five wells of the 96-well plates used in the cytotoxicity assay. This clarification has been added to the revised manuscript.
[C8] This is a similar work that should be discussed "Link: https://www.ncbi.nlm.nih.gov/pmc/articles/PMC7940945/"
[R8] In accordance with the reviewer's comment, the suggested study has been included and discussed in the Discussion section of the revised manuscript. This discussion can be found in lines 533-537.
Reviewer 3 Report
Comments and Suggestions for Authors
The paper entitled “Exploring cardiac exosomal RNAs of acute myocardial infarction” by Jung et al is well written although there are some queries which needs to be address before giving the acceptance. The manuscript should be well structured and in my opinion the authors are failed to do so. For example, in mice model they used acute inflammation model whereas, in the in-vitro model, they induced the hypoxic condition to determine the results, how we can corelate both? Why did authors not isolate the cells from the MI induced by acute inflammation model rather than separation of fibroblast and cardiomyocytes and then conduct the research? Moreover, please check the grammatical and syntax errors.
Comments on the Quality of English LanguageThe text have moderate grammatic and syntax errors
Author Response
Responses to the reviewer’s comments (Biomedicines-2844574)
We would like to thank the reviewer again for their careful reading and valuable comments that have really helped us improve the current revised version of our manuscript. We have addressed all the general and specific comments made by the referee in order to improve the clarity of the manuscript. We agree with most of his/her propositions and updated our manuscript accordingly. Moreover, and in order to help the assessors in their reviewing process, we have marked all the new changes and corrections in the revised version in blue. A full point-by-point list of responses and explanations is provided below.
C: comments by the reviewer; R: responses by the authors
Reviewer #3:
The paper entitled “Exploring cardiac exosomal RNAs of acute myocardial infarction” by Jung et al is well written although there are some queries which needs to be address before giving the acceptance.
[C1] The manuscript should be well structured and in my opinion the authors are failed to do so. For example, in mice model they used acute inflammation model whereas, in the in-vitro model, they induced the hypoxic condition to determine the results, how we can corelate both?
[R1] In response to the concerns raised, we have added a clarification in the manuscript. Myocardial infarction (MI) is characterized by myocardial ischemia and hypoxia due to coronary artery occlusion, which leads to inflammation and apoptosis [1]. Acute myocardial infarction (AMI) specifically results in hypoxic stress to cardiomyocytes [2]. Consequently, primary cardiac cells or cardiac cell lines exposed to hypoxic stress are commonly used as in vitro models of MI [3-5]. In this study, we employed primary cardiomyocytes and fibroblasts, the predominant cell types in the mouse heart, to simulate MI conditions in vitro using hypoxia. This rationale and corresponding references have been incorporated into the revised manuscript, particularly in lines 492-495, to better explain the correlation between the in vivo and in vitro models used.
[1] Jiang, K.; Kang, L.; Jian, A.; Zhao, Q. Development and Validation of a Diagnostic Model Based on Hypoxia-Related Genes in Myocardial Infarction. Int. Gen. Med. 2023, 29, 2111-2123.
[2] Datta Chaudhuri, R.; Banik, A.; Mandal, B.; Sarkar, S. Cardia-specific overexpression of HIF-1α during acute myocardial infarction ameliorates cardiomyocyte apoptosis via differential regulation of hypoxia-inducible pro-apoptotic and anti-oxidative genes. Biochem. Biophys. Res. Commun. 2021, 22, 100-108.
[3] Jin, H.; Yu, J. Lidocaine Protects H9c2 Cells from Hypoxia-induced Injury through Regulation of the MAPK/ERK/NFκB Signaling Pathway. Exp. Ther. Med. 2019, 18, 4125-4131.
[4] Cai, Y.; Li, Y. Upregulation of MiR-29b-3p Protects Cardiomyocytes from Hypoxia-induced Apoptosis by Targeting TRAF5. Cell. Mol. Biol. Lett. 2019, 11, 27.
[5] Pyo, J.O.; Nah, J.; Kim, H.J.; Chang, J.W.; Song, Y.W.; Yang D.K.; Jo, D.G.; Kim, H.R.; Chae, H.J.; Chae, S.W.; Hwang, S.Y.; Kim, S.J.; Kim, S.J.; Kim, H.J.; Cho, C.; Oh, C.G.; Park, W.J.; Jung, Y.K. Protection of Cardiomyocytes from Ischemic/hypoxia Cell Death via Drbp1 and pMe2GlyDH in Cardio-specific ARC transgenic mice. J. Biol. Chem. 2008, 7, 30707-30714.
[C2] Why did authors not isolate the cells from the MI induced by acute inflammation model rather than separation of fibroblast and cardiomyocytes and then conduct the research?
[R2] The nature of myocardial infarction (MI) involves the death of cardiac myocytes due to prolonged ischemia. In cases of acute myocardial ischemia extending beyond 20 minutes, there is a progressive 'wave front' of cardiomyocyte death, starting from the subendocardium and moving transmurally towards the pericardium [1]. Due to this progression, isolating a sufficient number of viable cells from heart tissues affected by MI for in vitro experiments is challenging. Consequently, primary cardiomyocytes and fibroblasts from neonatal mouse hearts were utilized for the in vitro experiments in our study, as they offer a more reliable and controlled method to investigate the effects of MI.
[1] Hashmi, M.; AI-Salam, S. Acute Myocardial Infarction and Myocardial Ischemia-reperfusion Injury: A Comparison. Int. Clin. Exp. Pathol. 2015, 1, 8786-8796.
[C2] Moreover, please check the grammatical and syntax errors.
[R3] Thank you for your insightful comments. We have thoroughly reviewed the manuscript for grammatical and syntax errors, as suggested, and made the necessary corrections throughout the revised manuscript.